# Spatial–Temporal Patterns and Propagation Dynamics of Ecological Drought in the North China Plain

Zezhong Zhang [1], Hexin Lai [1], Fei Wang [1,2,*], Kai Feng [1], Qingqing Qi [1] and Yanbin Li [1]

1   School of Water Conservancy, North China University of Water Resources and Electric Power, Zhengzhou 450046, China; zhangzezhong@ncwu.edu.cn (Z.Z.); x201910102868@stu.ncwu.edu.cn (H.L.); fengkai@ncwu.edu.cn (K.F.); qiqingqing@ncwu.edu.cn (Q.Q.); liyanbin@ncwu.edu.cn (Y.L.)
2   Yinshanbeilu Grassland Eco-Hydrology National Observation and Research Station, China Institute of Water Resources and Hydropower Research, Beijing 100038, China
*   Correspondence: wangfei@ncwu.edu.cn; Tel.: +86-1-823-692-7593

**Abstract:** With the increasingly prominent ecological environmental problems, the ecological drought phenomenon will become the focus of drought research. The spatial–temporal pattern of ecological drought and its complicated propagation dynamics are still unclear in the North China Plain (NCP). In this study, the spatio–temporal patterns and gridded trend characteristics of ecological drought were investigated from 1999 to 2019 in the NCP. Subsequently, the propagation dynamics from meteorological to ecological drought were identified for the study period. The results indicated that: (1) the ecological drought showed a downward trend from 1999 to 2019 in the NCP, with a 4.2-year and 7-year period on the inter-annual scale; (2) the most serious ecological drought occurred in the year of 2002, with an average monthly vegetation condition index (VCI) ranging from 0.17 (in December) to 0.59 (in January); (3) ecological droughts were decreasing in each month and season at the grid scale; (4) the propagation time from meteorological to ecological drought was 2.67 months in winter and 1.33 months in summer, which is helpful for predicting the occurrence of ecological drought. This study sheds new viewpoints into the spatial–temporal patterns and propagation dynamics of ecological drought in the NCP, which can also be applied in Northeast China.

**Keywords:** ecological drought; meteorological drought; propagation dynamics; multivariate cross wavelet transform technology; North China Plain (NCP)





## 1. Introduction

In recent years, with the intensification of climate change, the problem of ecological drought has gradually attracted people's extensive attention [1,2]. Serious and prolonged ecological drought will not only increase the vulnerability of the ecosystem, raise the mortality of trees and change the state of the ecosystem, but also bring immeasurable losses to humans and nature [3–5]. Ecological drought will cause the reduction of water area, the decline of vegetation activity and the change in pollutant formation and migration. Under extreme drought conditions, it even leads to serious ecological consequences, such as the change of spatial distribution characteristics of aquatic organisms, the decline of reproductive capacity and the reduction of population size [6]. Ecological drought is a periodic insufficient water supply caused by natural conditions or human activities, which leads to the change of hydrological and meteorological conditions for the normal growth and development of vegetation. Crausbay et al. defined ecological drought as an intermittent shortage of water supply, resulting in the vulnerability of the ecosystem exceeding its threshold [2]. Ulteriorly, the conceptual framework of ecological drought is composed of two dimensions: vulnerability components and the continuous unity of human-natural factors, which is helpful for ecological drought researchers and decision makers to understand. With the increasingly prominent eco-environmental problems, ecological drought will become a hotspot issue of drought research in the future.

Many scholars have carried out different studies on spatio-temporal drought pattern modeling. Kwon et al. explored the spatio-temporal drought patterns of multiple drought indices based on precipitation and soil moisture and observed persistent droughts with higher severity in the northern part of South Korea [7]. Diaz et al. proposed an approach to characterize spatio-temporal drought dynamics [8]. Bahmani et al. depicted spatial variability of drought-prone areas using a geographically weighted regression model in the northwest of Iran [9]. Li et al. identified spatio-temporal drought events based on multi-source remote sensing data in eastern China [10]. Afshar et al. used linear relationships between the meteorological and soil moisture drought indices to discuss their consistency [11].

With the aggravation of drought intensity and frequency, more ecologists began to pay attention to the ecological effects of drought [12]. For example, Hou et al. proposed an ecological drought index, based on the principle of minimum ecological water level and water balance, to evaluate ecological drought in the Hulun wetland of China [13]. Jiang et al. compared some widely used drought indices and constructed the standardized ecological water deficit index to quantitatively monitor the severity of ecological drought in Northwest China [5]. Kim et al. used a nonparametric kernel density method to assess the water quality risk in the Korean Nakdong River basin [14]. In addition, California's drought during 2012–2015, with deficient rainfall and exceptional warmth, led to the death of a large area of coniferous forest [15]. Since the 21st century, the severe drought has led to a nearly 50% reduction in forest land area compared with that in the 1950s, resulting in ecological environment deterioration and vegetation degradation in the Qilian Mountains [16].

Ecological drought evaluation is very important for improving ecological environment quality, restoring ecosystem function and controlling ecological degradation, which is also an effective approach to promote the coordinated development of regional society and environment. Remote sensing satellites play a great role in investigating drought, which can provide valuable information in thermal infrared, mid-infrared, near-infrared, and microwave bands [17–19]. Nowadays, the spectral reflection information-based remote sensing drought indices can reflect the response characteristics of vegetation growth to drought, quantitatively determine the shortage of vegetation state to water supply, and are suitable for large-scale drought investigation and evaluation [20–24]. During the drought period, with the increase of evaporation potential, the over-exploitation of groundwater makes the vegetation dependent on groundwater suffer drought stress and leads to ecological drought occurrence. Noticeably, water stress caused by drought can hinder the physiological growth of vegetation, and affect the seasonal normal growth and development of forest trees, farmland crops and herbaceous plants, resulting in the reduction of terrestrial vegetation coverage and greenness, and the deterioration of regional ecological environment [25–28]. Additionally, the vegetation index based on remote sensing is closely related to green leaf biomass, leaf area index, plant photosynthetic capacity, total dry matter accumulation and net primary productivity, which can accurately reflect the growth status, biomass, coverage rate and photosynthesis intensity of vegetation, and can also provide theoretical support for the disaster events on the ecological environment [29–31].

The research on the propagation dynamics among different types of drought can effectively analyze the occurrence pattern and formation mechanism of drought, which has important theoretical significance for improving the ability of drought early warning and prevention [32–36]. The relationship between meteorological and hydrological drought may be highly affected by local watershed characteristics and climate conditions, and the propagation time may be different in each season [37]. Furthermore, El Niño-Southern Oscillation (ENSO) and Arctic Oscillation (AO) have a strong correlation with regional actual evaporation [38]. Due to various climatic characteristics in different regions, the propagation relationship from meteorological to agricultural drought was stronger than that from agricultural to hydrological drought [39]. However, the current drought research pays insufficient attention to the driving factors and relevant mechanisms of drought propagation

and has not revealed the dynamic propagation characteristics from meteorological drought to ecological drought.

Therefore, the remote sensing-based vegetation condition index (VCI) was selected as an ecological drought index, and the spatial–temporal patterns of ecological drought were identified during 1999–2019 in the North China Plain. The main aims and goals of the current study are (1) to reveal the temporal evolutions and spatial patterns of ecological drought; (2) to identify the gridded ecological drought trend characteristics; (3) to explore the propagation dynamics from meteorological to ecological drought; and (4) to determine the dynamic relationships between teleconnection factors and ecological drought quantitatively.

## 2. Study Region

As a typical alluvial plain and the most populous plain in China, North China Plain (NCP), also called the Huang-Huai-Hai Plain, is located between 110°–122° E and 31°–43° N. Most areas of the NCP are located in temperate and subtropical monsoon climate areas, with an average annual precipitation of 400–600 mm [36]. The temporal and spatial distribution of water resources is uneven, and the per capita water resources is only 456 m$^3$/year. The northern part of the NCP is the Yanshan Mountains, the southern part is the Yellow River Basin, the western part is the Taihang Mountains and the Qinling Mountains, and the eastern part is the Yellow Sea and the Bohai Sea [40]. Based on geographical location and administrative region, there are five sub-zones in the NCP (Figure 1), i.e., Bei Jing (BJ), Tian Jin (TJ), Shan Dong (SD), He Bei (HB) and He Nan (HN). The details of the above five sub-zones are shown in Table 1.

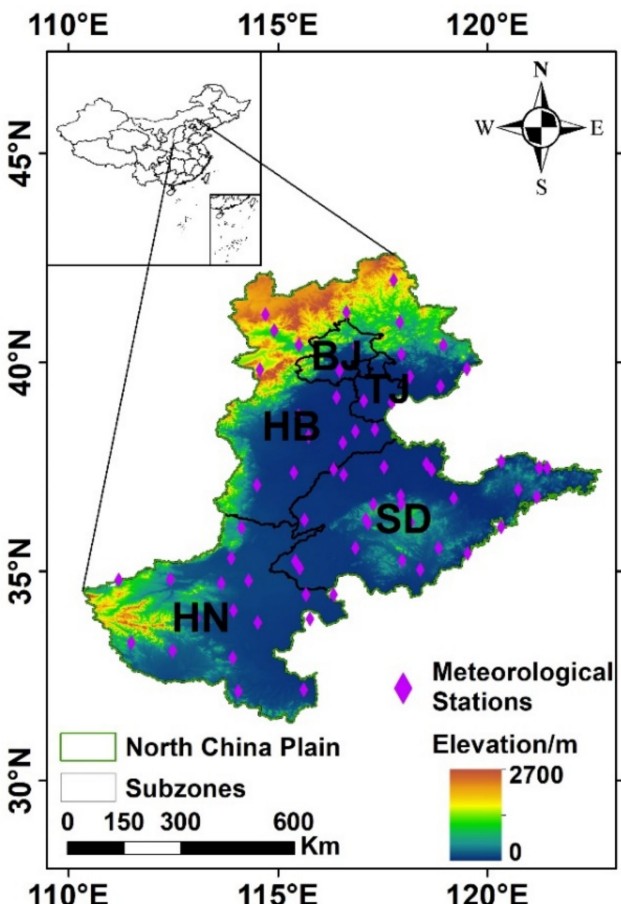

**Figure 1.** Administrative sub-zones and meteorological stations located in the NCP.

**Table 1.** List of regions in this study.

| Region | Abbreviation | Area ($10^4$ km$^2$) | Number of Stations |
|---|---|---|---|
| Bei Jing | BJ | 1.73 | 2 |
| Tian Jin | TJ | 1.21 | 2 |
| Shan Dong | SD | 15.39 | 27 |
| He Bei | HB | 19.64 | 20 |
| He Nan | HN | 16.14 | 16 |
| North China Plain | NCP | 54.11 | 67 |

## 3. Materials and Methods

### 3.1. Materials

#### 3.1.1. Remote Sensing Satellite Data

The monthly normalized difference vegetation index (NDVI) dataset was selected from SPOT/VEGETATION satellite covering 1999–2019 with a spatial resolution of 1 km, which had been post-processed by geometric correction, radiometric calibration and noise reduction (https://www.resdc.cn/, accessed on 14 February 2019). As a standardized vegetation index, NDVI compares the pigment absorption rate of chlorophyll in the red band with the high reflectivity of plants, in order to generate an image for quantitatively exhibiting vegetation biomass [41,42]. Using ArcGIS 10.3 software for format transformation, the monthly, seasonal and annual NDVI datasets were finally clipped and composed.

#### 3.1.2. Meteorological Station Data

The meteorological station data were obtained from the National Meteorological Science Data Center in China for 67 meteorological stations during 1999–2019 (http://data.cma.cn/, accessed on 14 February 2019). These data have been strictly examined through homogeneity test and extreme value test with satisfactory quality requirements. Moreover, the number of meteorological stations in BJ, TJ, SD, HB and HN are 2, 2, 27, 20 and 16, respectively.

#### 3.1.3. Teleconnection Factors

In this study, eight large-scale atmospheric circulation factors affecting global and regional climate were selected from 1999 to 2019, including the ENSO, Pacific Decadal Oscillation (PDO), AO, North Atlantic Oscillation (NAO), Atlantic Multi-decadal Oscillation (AMO), Dipole Mode Index (DMI), North Pacific Index (NPI) and Pacific-North American (PNA). These teleconnection data were derived from the National Oceanic and Atmospheric Administration (https://psl.noaa.gov/gcos_wgsp/Timeseries/, accessed on 14 February 2019).

#### 3.1.4. Digital Elevation Model Data

Based on the latest Shuttle Radar Topography Mission (SRTM) V4.1 dataset, the digital elevation model (DEM) data was resampled and generated using WGS84 ellipsoidal projection, which was used for ecological environment investigation and had the advantages of free access and strong actuality.

### 3.2. Methods

#### 3.2.1. Vegetation Condition Index (VCI)

As a drought indicator monitored by remote sensing satellite, VCI can reflect the comprehensive information on vegetation growth state and the ecological response of terrestrial vegetation to drought [43,44]. Additionally, VCI is a metric of crop growth and surface vegetation coverage, which can reduce the impact of different soil conditions, geographical locations and ecosystems on NDVI. The smaller VCI value indicates worse vegetation growth status and a higher degree of drought. The detailed calculation process of VCI can be referred to [44].

### 3.2.2. Standardized Precipitation Evapotranspiration Index (SPEI)

Standardized precipitation evapotranspiration index (SPEI) preserves the simple calculation procedures of standardized precipitation index and has spatial comparability, which is widely used in meteorological drought investigation [45,46]. Meanwhile, SPEI combines two factors of precipitation and evapotranspiration, which makes it more suitable for evaluating the drought evolution process under the background of climate change [46].

### 3.2.3. Extreme-Point Symmetric Mode Decomposition (ESMD)

Extreme-point symmetric mode decomposition (ESMD) method is a new development of the empirical mode decomposition method, and it is also one of the methods to extract the nonlinear change trend of the signal [47,48]. One advantage of ESMD is that it can define the optimal global moving average through an adaptive method, and the other advantage is that it can directly determine the instantaneous amplitude. In this study, ESMD was adopted to reveal the temporal evolutions of VCI. Details about the ESMD method can be referred to [47].

### 3.2.4. Gridded Trend Identification Method

The modified Mann–Kendall (MMK) trend test method is employed to capture the trends of hydro-meteorological time series. The MMK method can improve the test ability of the traditional Mann–Kendall method and eliminate the autocorrelation components in the original sequence, and it is an effective nonparametric method to identify the change trend of time series. In this study, we improved the MMK method to obtain the drought trend characteristics at the grid scale. More details on the MMK method can be found in [49].

### 3.2.5. Rescaled Range (R/S) Analysis

Proposed by British scholar Hurst in 1951, the rescaled range (R/S) analysis method is a fractal structure processing method for distinguishing time series, which is used in determining the persistence or anti-persistence intensity of time series change trend [50,51]. In the R/S analysis method, a Hurst index > 0.50 indicates that the future trend is consistent with the historical trend, with stronger correlation and trend persistence as the index is close to 1. The R/S analysis method was used to identify the trend characteristics of ecological drought in the future. The primary calculation procedures are clarified in [51].

### 3.2.6. Cross Wavelet Transform Technology

In view of the non-normal distribution of many geophysical time series, the cross wavelet transform technology can be used to reveal the common power and relative phase in time-frequency space. The cross wavelet transform can find out the regions with the same higher power, while wavelet coherence can reveal the correlation of this cross transform in both time and frequency domains [52,53]. The cross wavelet method was used to explore the dynamic relationships between ecological drought and teleconnection factors, so as to reveal the driving effect of teleconnection factors on ecological drought in the NCP.

In addition, an overall methodology flow chart is shown in Figure 2.

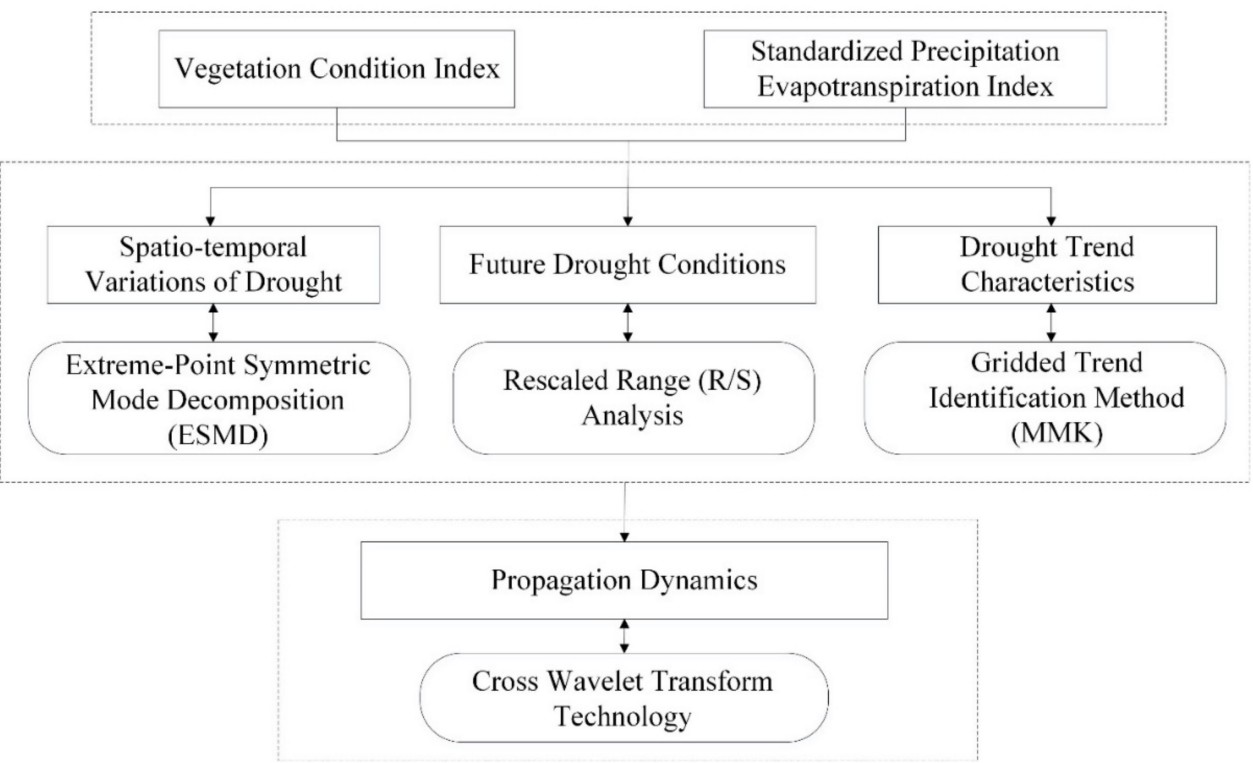

**Figure 2.** The methodology flow chart in this study.

## 4. Results

### 4.1. Temporal Evolutions of Ecological Drought

As shown in Figure 3, VCI time series can be decomposed based on ESMD during 1999–2019 in the NCP. When the optimal screening times reached 21 times, the ESMD-based trend item R (R = 0.34–0.58) can reflect the overall fluctuation characteristics of the original VCI sequence. Two intrinsic mode function (IMF) components and one trend item R can be obtained based on ESMD. Additionally, the sum of trend item R and IMF components completely coincided with the original VCI time series, demonstrating that the ESMD method was credible and reliable. Based on fast Fourier transform, the average periods of IMF1 and IMF2 were 4.2 and 7 years, indicating that ecological drought had the periodic characteristics of 4.2-year and 7-year on the inter-annual scale in the NCP. Meanwhile, the variance contribution rates of trend item R, IMF1 and IMF2 were 76.67%, 21.63% and 1.71%, respectively. It could be concluded that the trend item R contributed to the temporal evolutions of ecological drought. Furthermore, the Hurst index of VCI sequence reached 0.53 (>0.50) calculated by R/S analysis, implying that the trend of ecological drought in the future is consistent with the original downward trend. The original VCI time series reached 0.61 and 0.37 in 2012 and 2002, respectively, with the trend item R presenting a consistent and processive upward trend from 1999 to 2019.

Similarly, the trend item R of seasonal and annual VCI can be obtained during 1999–2019 in different regions of the NCP based on ESMD (Figure 4). Obviously, the variation of VCI reflected by trend item R was different in each sub-region, with the most obvious upward trend occurring in HN, BJ, BJ and HN in each season. For the whole NCP, the trend item R showed a fluctuating upward trend, with the largest variation amplitude in summer (R = 0.33–0.65) and the smallest variation amplitude in winter (R = 0.37–0.54). From spring to winter, the R contribution rates in the NCP were 74.28%, 69.20%, 56.40% and 17.80%, respectively. In general, the trend items of VCI were increasing, indicating that the ecological drought showed a downward trend for the study period.

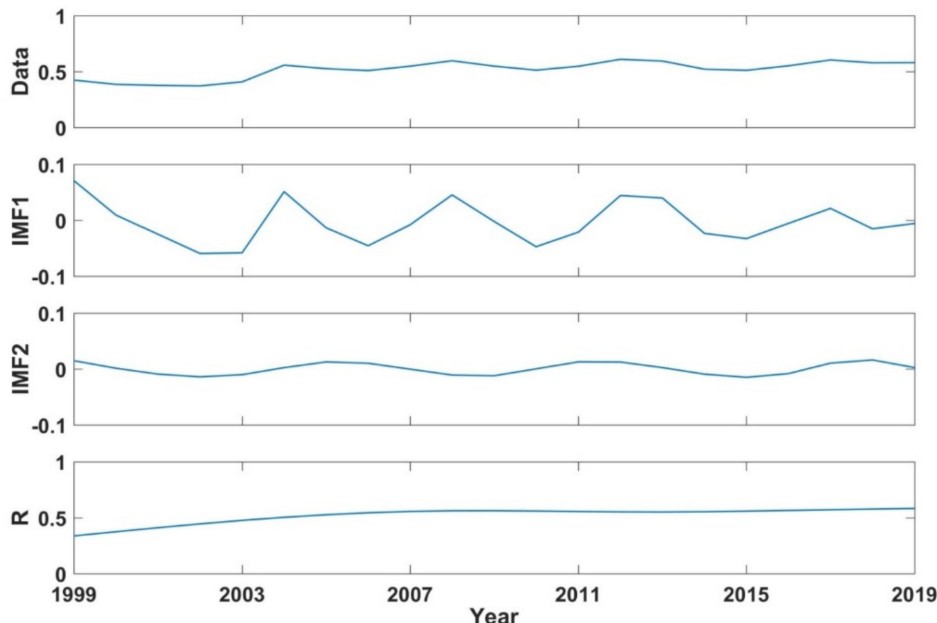

**Figure 3.** Two IMF components and one trend item R obtained by ESMD decomposition.

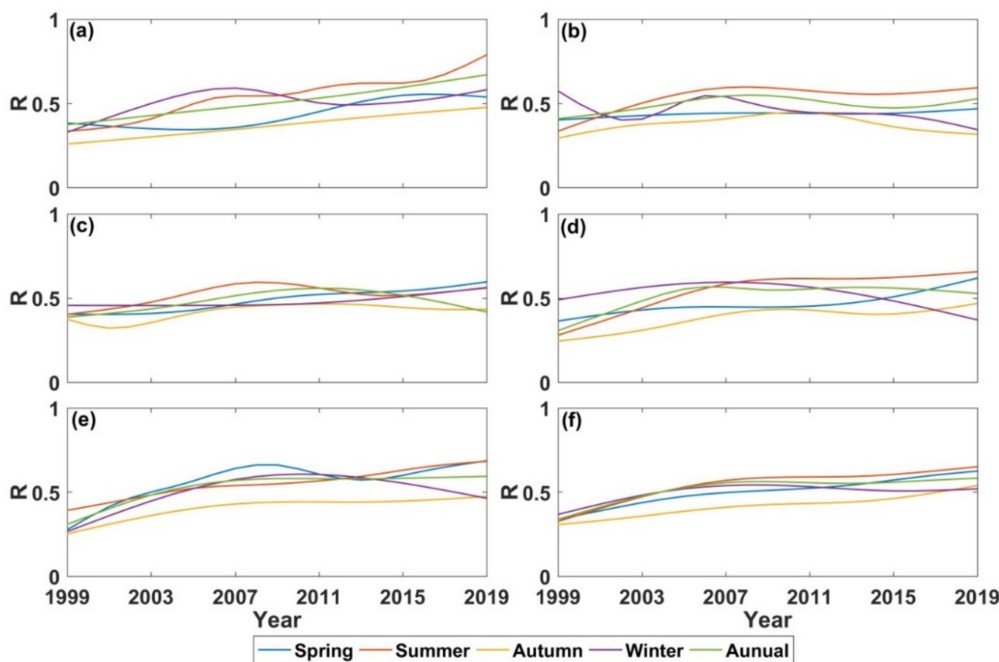

**Figure 4.** Trend item R of VCI during 1999–2019 in each region (**a**) BJ, (**b**) TJ, (**c**) SD, (**d**) HB, (**e**) HN and (**f**) NCP.

### 4.2. Spatial Patterns of Ecological Drought

Figure 5 depicts the spatial patterns of monthly and seasonal ecological drought in the NCP. In each month, the worst ecological drought with the minimum VCI values occurred in SD (0.48), SD (0.43), TJ (0.44), TJ (0.41), TJ (0.46), HN (0.45), BJ (0.53), BJ (0.54), BJ (0.54), HN (0.47), BJ (0.46) and TJ (0.47), respectively (Figure 5a–l). Additionally, based on average VCI values for all pixels, the monthly VCI ranged from 0.47 (in October) to 0.58 (in August). In each season, the most serious ecological drought with the minimum VCI appeared in TJ (0.44), SD (0.53), HB (0.35) and TJ (0.47), respectively (Figure 5m–p). In spring, summer, autumn and winter, the average VCI was 0.50, 0.54, 0.39 and 0.51, suggesting that the worst ecological drought occurred in autumn. Noteworthily, the average VCI was 0.50, 0.48, 0.49,

0.51 and 0.53 in BJ, TJ, SD, HB and HN, respectively. Therefore, TJ and SD were vulnerable to droughts.

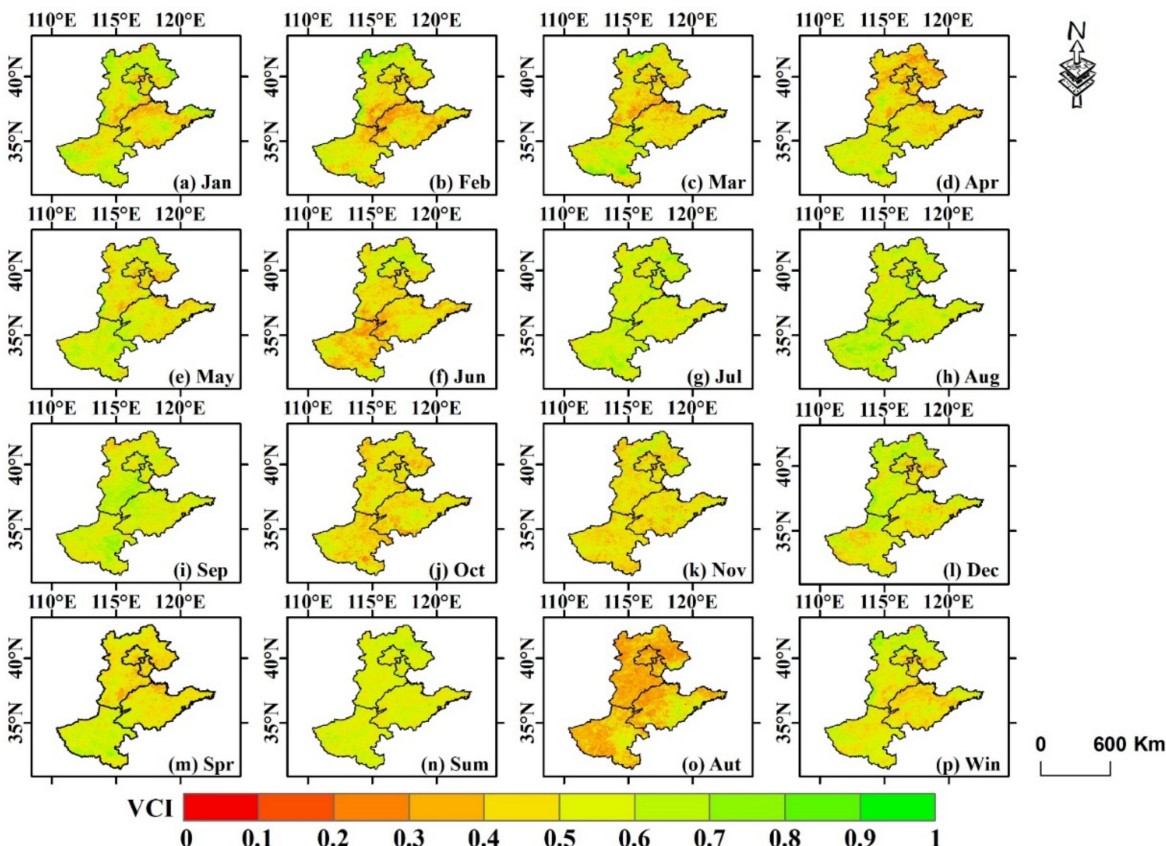

**Figure 5.** Spatial patterns of monthly and seasonal ecological drought in the NCP. (**a**–**p**) denote January, February, March, April, May, June, July, August, September, October, November, December, Spring, Summer, Autumn and Winter, respectively.

Since the minimum annual VCI occurred in the year 2002 from 1999 to 2019, we considered 2002 as a representative drought year for identifying the spatial distribution characteristics of ecological drought in the NCP (Figure 6). On a monthly scale, the maximum and minimum VCI occurred in January (0.59) and December (0.17), respectively (Table 2). The driest area occurred in SD, with an average monthly VCI value of 0.30. On a seasonal scale, the maximum and minimum average VCI occurred in spring (0.41) and winter (0.24), respectively. Summer drought was mainly concentrated in BJ, with an average VCI value of 0.29. Spring drought, autumn drought and winter drought were primarily concentrated in SD, with average VCI values of 0.35, 0.23 and 0.17, respectively. Apparently, the worst ecological drought appeared in SD in the year 2002.

**Table 2.** The maximum, minimum and mean values of VCI during 2002.

| VCI Value. | Jan | Feb | Mar | Apr | May | Jun | Jul | Aug | Sep | Oct | Nov | Dec | Spr | Sum | Aut | Win |
|---|---|---|---|---|---|---|---|---|---|---|---|---|---|---|---|---|
| Maximum | 0.64 | 0.63 | 0.71 | 0.49 | 0.37 | 0.26 | 0.50 | 0.57 | 0.64 | 0.40 | 0.41 | 0.33 | 0.45 | 0.42 | 0.32 | 0.29 |
| Minimum | 0.50 | 0.42 | 0.43 | 0.35 | 0.17 | 0.14 | 0.35 | 0.31 | 0.27 | 0.20 | 0.17 | 0.05 | 0.35 | 0.29 | 0.23 | 0.17 |
| Mean | 0.59 | 0.55 | 0.58 | 0.40 | 0.24 | 0.23 | 0.40 | 0.39 | 0.39 | 0.27 | 0.25 | 0.17 | 0.41 | 0.34 | 0.25 | 0.24 |

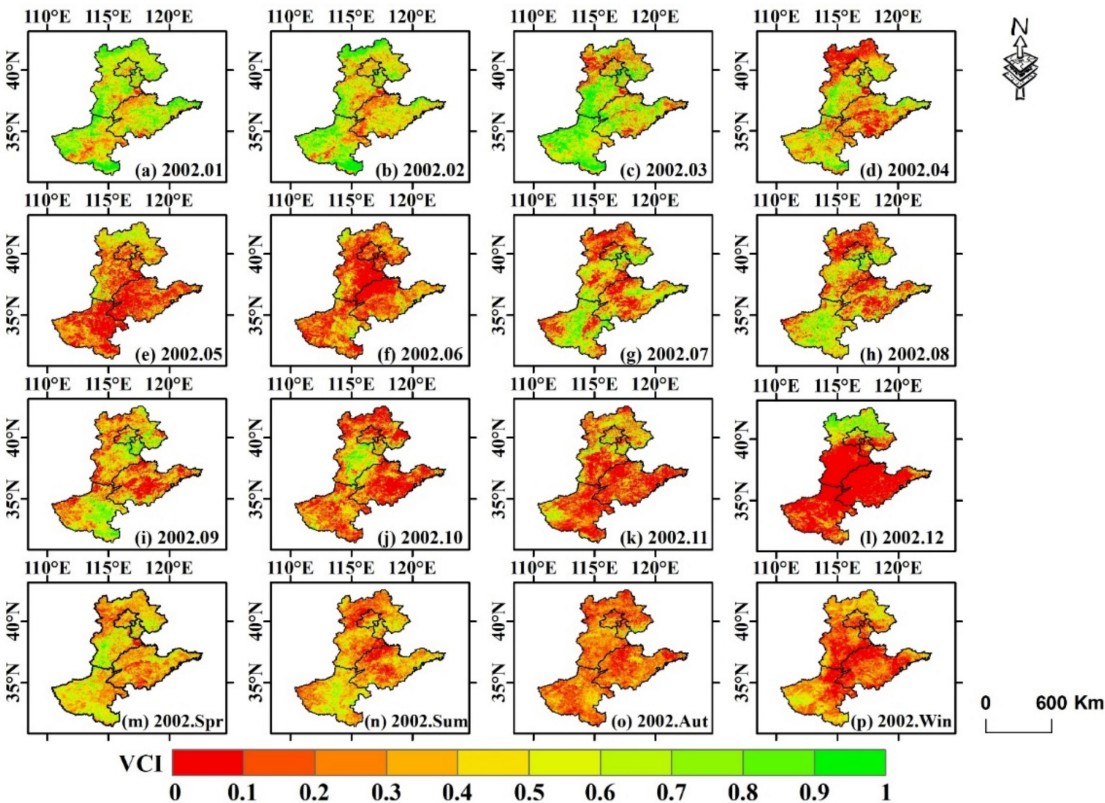

**Figure 6.** Spatial patterns of ecological drought in the year 2002 for the study area. (**a–p**) denote January, February, March, April, May, June, July, August, September, October, November, December, Spring, Summer, Autumn and Winter in 2002, respectively.

*4.3. Gridded Ecological Drought Trend Identification*

Figure 7 depicts the gridded trend identifications of VCI at the grid scale during 1999–2019 in the NCP. The trend characteristic $Z_s$ values of VCI for the study period are signified in Figure 8. The average trend characteristic $Z_s$ values were 0.73, 0.05, 0.47, 1.27, 1.93, 2.00, 1.03, 1.51, 1.48, 1.12, 0.70 and 0.46, respectively (Figure 7a–l). On the whole, ecological drought showed a downward trend from January to December. The average $Z_s$ values were greater than 0 in BJ, HB and HN for all months, indicating that ecological droughts were alleviating for each month in these three sub-zones. The area percentage with an upward drought trend ranged from 11.6% (in June) to 47.2% (in December). In addition, the largest (10.07%) and smallest (0.96%) area percentage ($p < 0.01$) occurred in March and June. The average $Z_s$ values were 1.60, 1.88, 1.46 and 0.64 from spring to winter, implying that ecological droughts were decreasing in each season for the study area (Figure 7m–p). The average $Z_s$ values were greater than 0 in spring, summer and autumn for all sub-regions, indicating that ecological droughts were decreasing for each sub-zone in these seasons. Moreover, for each season, the area percentage with an upward drought trend was 24.2%, 17.6%, 20.2% and 33.0%, respectively. Meanwhile, the largest (7.82%) and smallest (3.24%) area percentage ($p < 0.01$) occurred in spring and autumn. In BJ, the decreasing ecological drought trend was significant in May, June and summer at $p < 0.01$, and was significant in August, September, October, spring and autumn at $p < 0.05$. In HB, the significant downward trend of ecological drought mainly occurred in May, June, August and summer ($p < 0.05$). Furthermore, in HN, the significant downward trend of ecological drought mainly occurred in May, June, spring and summer ($p < 0.05$). Additionally, the increasing ecological drought mainly appeared in TJ and SD, which was consistent with the results of ecological drought-prone areas (Figure 5).

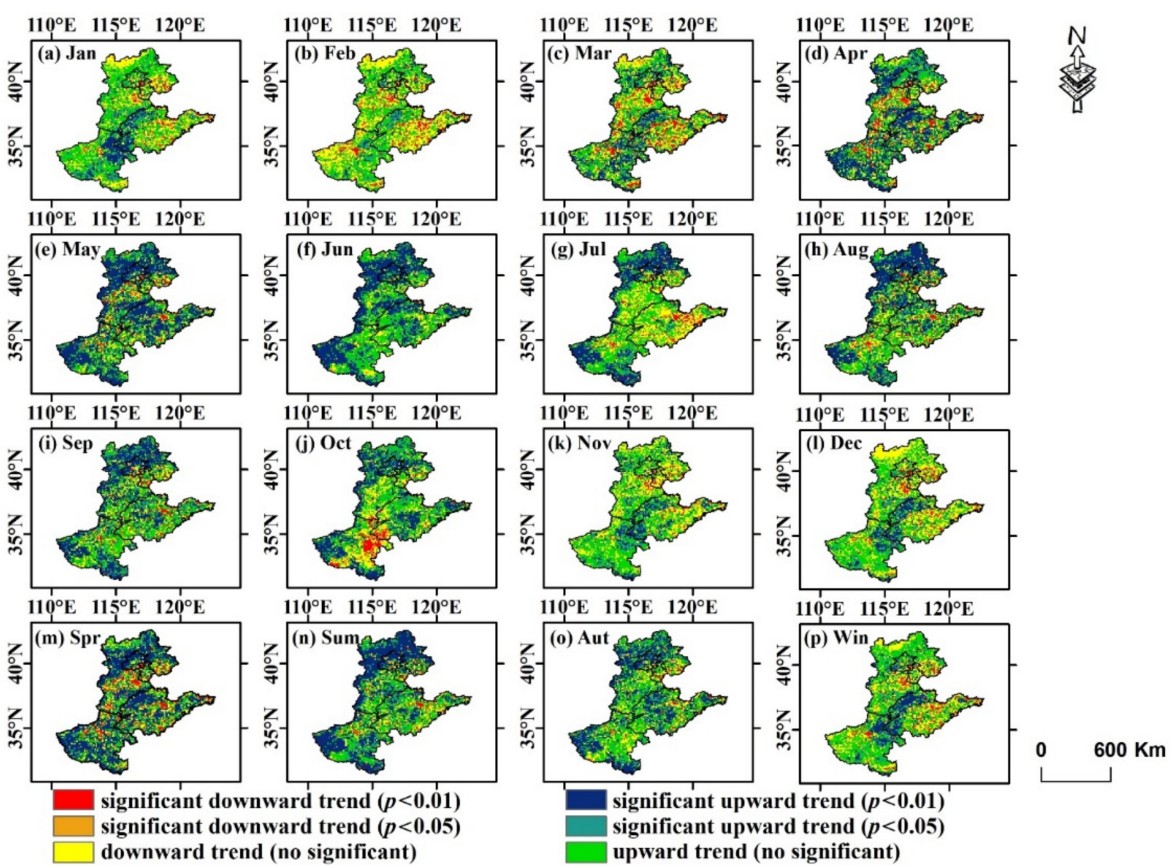

**Figure 7.** Gridded VCI trend identification from 1999 to 2019 in the NCP. (**a**–**p**) denote January, February, March, April, May, June, July, August, September, October, November, December, Spring, Summer, Autumn and Winter, respectively.

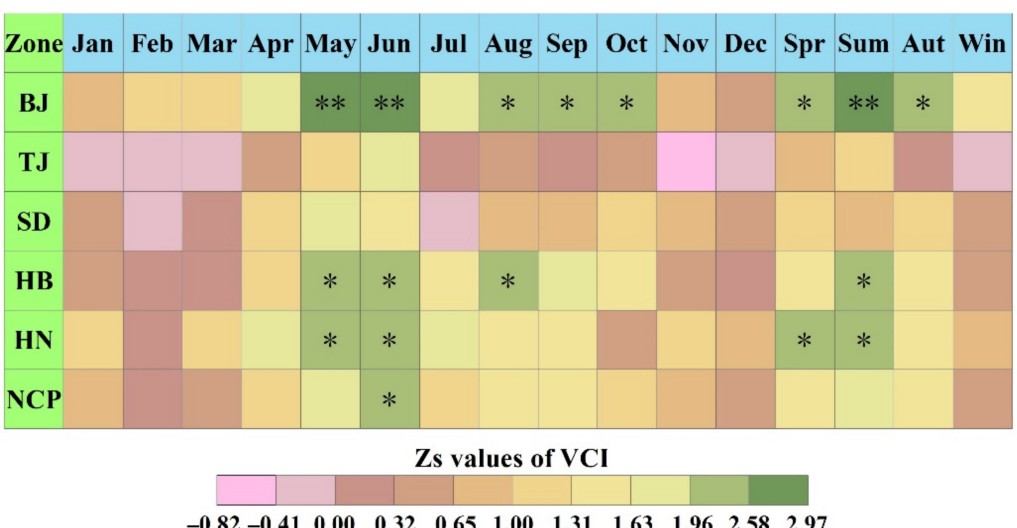

**Figure 8.** $Z_s$ values of VCI during 1999–2019 in the NCP. "*" and "**" denote significant trend at $p < 0.05$ and $p < 0.01$, respectively.

### 4.4. Propagation Dynamics from Meteorological to Ecological Drought

Previous studies [54,55] pointed out that the delayed time of vegetation to climate may vary depending on different climatic conditions. As presented in Figure 9, we analyzed the correlation coefficient (*r*) between VCI in each month and the meteorological drought index SPEI in the previous 0–3 months and regarded the time scale corresponding to the

largest *r* value as the propagation time [36,39]. In the NCP, the propagation time of 1 month occurred in May, June, August and September, with *r* values between VCI and SPEI of 0.60, 0.68, 0.70 and 0.56, respectively. Additionally, the propagation time of 2 months occurred in February, March, July and October, with *r* values of 0.55, 0.61, 0.67 and 0.61, respectively. Furthermore, the propagation time of 3 months occurred in January, April, November and December, with *r* values of 0.58, 0.62, 0.63 and 0.60, respectively. Meanwhile, the proportion of *r* values was 54.17% and 87.50% at $p < 0.01$ and $p < 0.05$. To sum up, we can conclude that the propagation time was longer in winter (2.67 months) with an average *r* value of 0.58, and shorter in summer (1.33 months) with an average *r* value of 0.68.

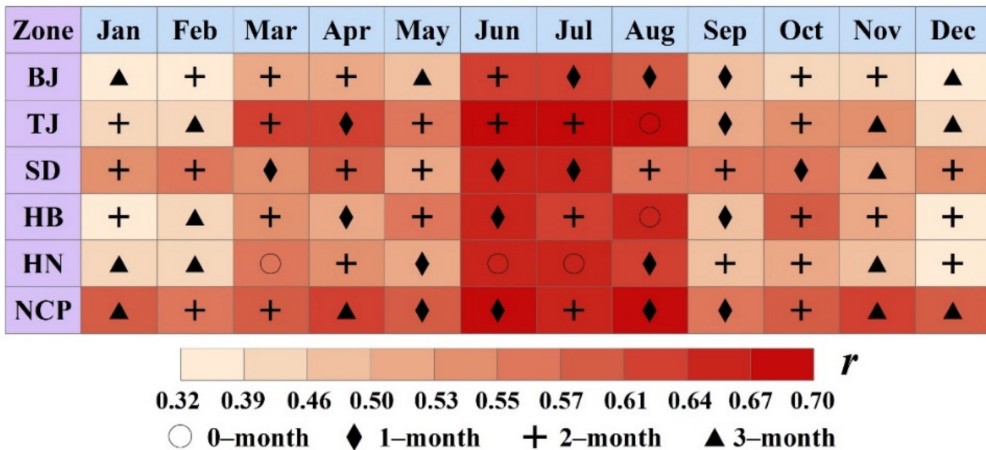

**Figure 9.** The propagation dynamics from meteorological to ecological drought in the NCP.

## 5. Discussion

### *5.1. Dynamic Relationships between Ecological Drought and Teleconnection Factors*

Relevant studies have demonstrated that the change of large-scale teleconnection factors played vital roles in the generation mechanism of drought [56–60]. Therefore, we used the cross wavelet transform technology to explore the dynamic linkages between teleconnection factors and ecological drought in the NCP. In the process of cross wavelet transform, the arrow pointing left indicates that the relative phase is inconsistent with a negative correlation between two variables. Conversely, the arrow pointing right denotes that the relative phase is consistent with a positive correlation. Figure 10a shows that ENSO is positively correlated with VCI in 8–10 months in 2016. Figure 10b indicates that PDO is positively correlated with VCI in 16–32 months (2005–2008), 12–14 months (2012–2013) and 12–16 months (2016–2018), and negatively correlated with VCI in 14–24 months (2009–2012). There is an obvious positive correlation between AO and VCI, on a scale of 10–16 months in 2012–2018 (Figure 10c). Furthermore, there is an obvious negative correlation between NAO and VCI, on a scale of 8–12 months in 2016–2017 (Figure 10d). Meanwhile, Figure 10e–h illustrated the relationships between the other teleconnection factors and VCI. Additionally, there is a period of 1–8 months between teleconnection factors and VCI during 1999–2019. In general, PDO has the highest dependence on ecological drought in the NCP.

In addition, we adopted wavelet coherence to reveal the mutual features between teleconnection factors and ecological drought in the low-energy areas for the study period in the NCP (Figure 11). Figure 11a indicates that ENSO has a negative correlation with VCI in 12–16 months (2000–2002), and a positive correlation with VCI in 8–12 months (2007–2010). Figure 11b depicts that PDO has four prominently negative correlations with VCI in 14–16 months (2003–2005), 14–24 months (2009–2011), 28–32 months (2010–2011) and 40–64 months (2007–2014), and a positive correlation with VCI in 8–14 months in 2016. Figure 11c shows that AO has a statistically significant positive correlation with VCI in 12–32 months (2001–2016). Meanwhile, NAO, AMO, DMI, NPI and PNA also have strong impacts on ecological drought (Figure 11d–h).

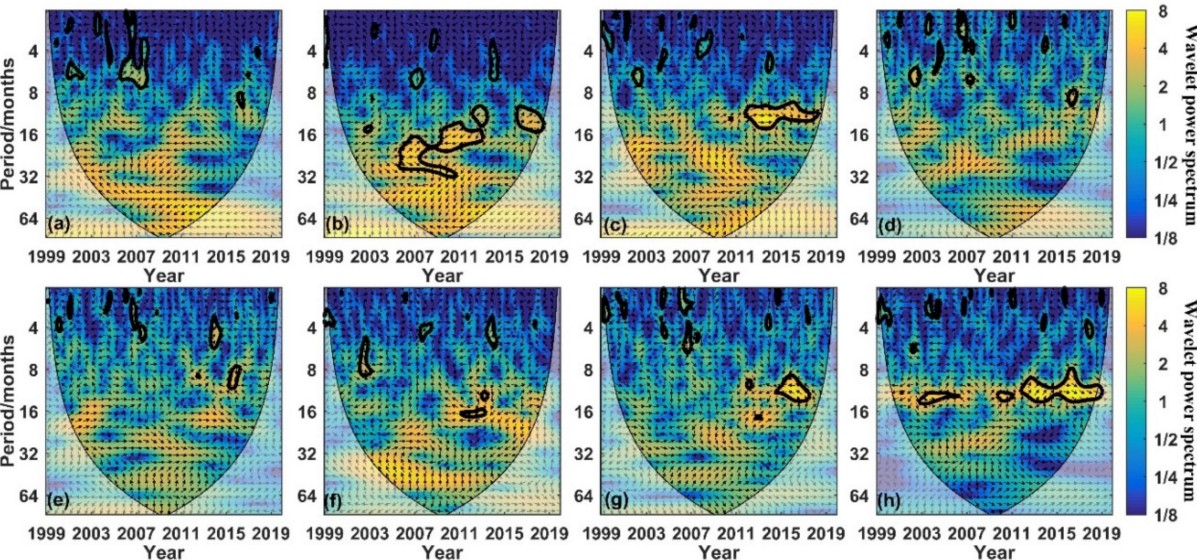

**Figure 10.** The cross wavelet transforms between the VCI and (**a**) ENSO, (**b**) PDO, (**c**) AO, (**d**) NAO, (**e**) AMO, (**f**) DMI, (**g**) NPI and (**h**) PNA data during 1999–2019 in the NCP.

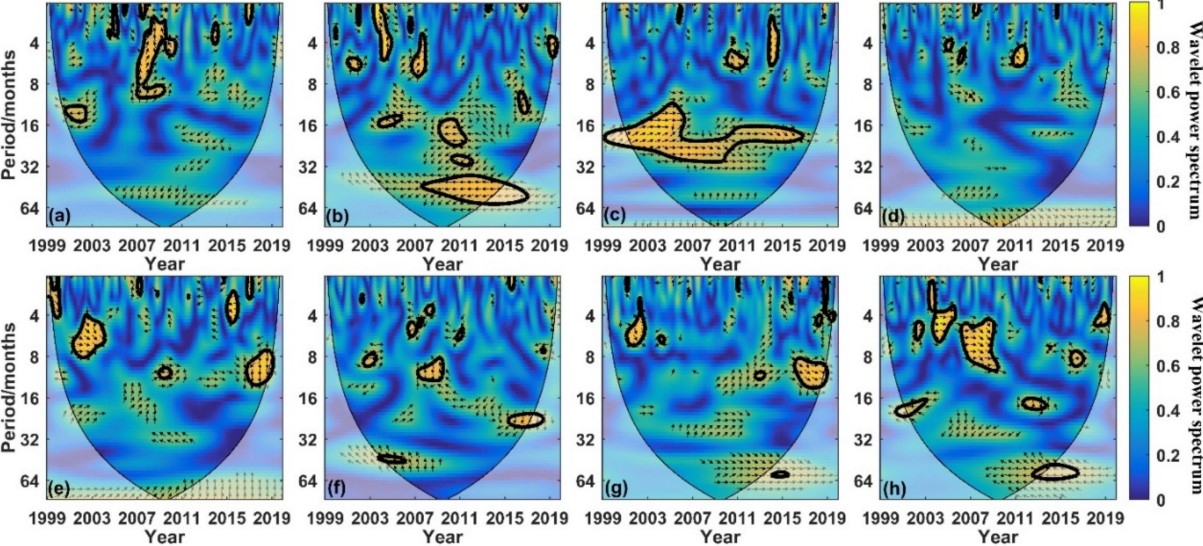

**Figure 11.** The wavelet coherence between the VCI and (**a**) ENSO, (**b**) PDO, (**c**) AO, (**d**) NAO, (**e**) AMO, (**f**) DMI, (**g**) NPI and (**h**) PNA data during 1999–2019 in the NCP.

### 5.2. Possible Dominant Influence Factors

The variation of precipitation patterns and the rise of global temperature are obviously driving the ecological drought evolutions [2,5]. In recent years, the ecological drought showed a slowing trend in the NCP. In particular, the most obvious ecological drought mitigation trend occurred in BJ, and the ecological drought in this region showed a decreasing trend in summer. Beijing is the capital and central city of China, and it is also the political and cultural center of China [61]. The policy and decision making management proposed by ecological departments such as soil erosion control and desertification control have been carried out in the NCP, resulting in the rising vegetation coverage, expanding agricultural cultivation scope, improving grain yield and slowing the ecological drought trend [62,63]. Meteorological drought caused by climate factor variation is also one major driving factor, which can reflect the change in climate parameters [64]. Since meteorological drought caused by deficient precipitation will cause the reduction of moisture content and vegetation greenness only when it reaches a certain degree, ecological drought has a

delayed response on meteorological drought [65]. Actually, ecological drought is closely related to other drought types through water–energy exchange on a regional scale. In addition, topographic conditions and anthropogenic activities (e.g., tree planting, grain for green, irrigation and crop management) will also affect the propagation process of drought [34,38,66].

Previous studies [67–70] have demonstrated that drought is sensitive to the change of large-scale circulation factors, and the relationship between teleconnection factors and drought has become a widespread topic for environmental and meteorological departments. PDO is a strong periodic marine atmospheric climate change model with a 10-year periodic scale, and it impacts the inter-decadal change of Asian climate by affecting the coastal and continental surface air temperature [71,72]. In this paper, it is recognized that PDO has the most obvious effect on ecological drought, which is a new discovery that can well explain the variations of ecological drought in the NCP (Figure 10). Thus, PDO can be used as an input factor to improve the prediction ability of ecological drought.

*5.3. Uncertainties*

There are several uncertainties that need to be clarified in this study. Firstly, due to the cloud blocking and adverse weather conditions, there are some uncertainties in remote sensing satellite datasets, which has been a universally existing problem in the use of remote sensing images [73–75]. Nonetheless, the products we used were re-processed through noise reduction and filtration with satisfying data quality. Secondly, other vegetation indices can also represent the physiological and ecological characteristics of vegetation, such as the vegetation health index (VHI), which combines the information on land surface temperature (LST) and vegetation status [76,77]. As determining and quantifying the weight of the two components of VHI is still a challenging task [31], it is practicable and feasible to utilize VCI for investigating the eco-physiological response of vegetation to the ecosystem. However, despite several of the above uncertainties, our study identified the spatio-temporal patterns of ecological drought and provided evidence of an overall downward drought trend.

*5.4. Advantages and Limitations*

The feedback mechanism of ecological drought on other influencing factors was not clear, thus, we applied VCI to reflect the ecological response ability of vegetation under drought stress. As one of the important indicators of the ecological environment, VCI is affected by climate, hydrology, underlying surface conditions and anthropogenic factors [25,64,78]. Due to different administrative districts and climate types, it is necessary to divide the whole NCP into several sub-zones for detailed study [36,40]. ESMD has obvious advantages in drought analysis and can reveal the overall fluctuation characteristics [79]. Based on ESMD, the most obvious increasing trend of the VCI sequence appeared in BJ, which can also be concluded from Figure 8. In this study, the gridded drought trend identification indicated that the ecological drought was decreasing from 1999 to 2019 in the NCP, which could be seen from the temporal evolutions of ecological drought in Section 4.1. The finding is consistent with other studies [51,55,63,80,81], which investigated the growth situation of vegetation in different regions.

Additionally, the investigation of drought propagation dynamics can enable us to predict the emergence of drought, which is helpful for maintaining the harmonious development of the eco-economy and promoting the virtuous circulation of the ecosystem [33–35,82]. As we know, previous studies [83,84] have shown that the drought propagations were complex, with the drought responses varying in different regions. Furthermore, the drought propagation time had significant regional differences and seasonal characteristics (Figure 9), and it was longer in winter (2.67 months) and shorter in summer (1.33 months). Ding et al. concluded that the response of drought propagation was stronger in summer ($r$ = 0.6–0.7), which was similar to our results [39].

Although the approach proposed in this study was successfully applied to evaluate the evolution characteristics of ecological drought, it still has some limitations. The available soil moisture products can be combined with vegetation growth conditions, which can provide accurate vegetation information [24,25]. In addition, as a new type of drought, the connotation of ecological drought needs to be clarified, and the indicators of ecological drought also have lots of potential for development in the future [4,5].

## 6. Conclusions

In this study, the spatial–temporal patterns and propagation dynamics of ecological drought were identified during 1999–2019 in the NCP. The temporal evolutions, spatial patterns and gridded trend characteristics of ecological drought were investigated. Additionally, the propagation dynamics from meteorological to ecological drought were clarified. Furthermore, the dynamic relationships between teleconnection factors and ecological drought were explored using cross wavelet transform technology. From the results, major conclusions are given as follows:

(1) The ecological drought was decreasing from 1999 to 2019 in the NCP, with a 4.2 year and 7 year period. Notably, the worst ecological drought appeared in the year 2002. The smallest VCI value (0.37) was found in 2002, and the average monthly VCI ranged from 0.17 (in December) to 0.59 (in January).

(2) In spring, summer, autumn and winter, the most serious ecological drought with the minimum VCI occurred in TJ (0.44), SD (0.53), HB (0.35) and TJ (0.47), respectively. Furthermore, the two ecological drought-prone areas in the NCP were TJ and SD.

(3) On a monthly scale, the largest (10.07%) and smallest (0.96%) area percentage ($p < 0.01$) occurred in March and June. On a seasonal scale, the largest (7.82%) and smallest (3.24%) area percentage ($p < 0.01$) occurred in spring and autumn, respectively.

(4) The propagation dynamics from meteorological to ecological drought had significant regional differences and seasonal characteristics. On the whole, the propagation time was longer in winter (2.67 months) with an average $r$ value of 0.58, and shorter in summer (1.33 months) with an average $r$ value of 0.68.

In summary, this study thoroughly explored the dynamic variations of ecological drought on the temporal and spatial scale and revealed the gridded ecological drought trend characteristics. Due to the availability of the remote sensing satellite data sets, we only analyzed the data in the recent 21 years, and this is a limitation of this study. In the future, the comprehensive ecological drought index can be constructed based on multi-source remote sensing data, which can reflect various information. In this paper, although the NCP was selected as a case study, the research approach can also be applied in other regions.

**Author Contributions:** Conceptualization, F.W. and Z.Z.; data interpretation and methodology, H.L.; validation, K.F. and Y.L.; software, Q.Q.; original draft preparation, Z.Z.; funding acquisition, F.W., Y.L. and Z.Z. All authors have read and agreed to the published version of the manuscript.

**Funding:** This research was supported by Major Science and Technology Projects in Henan Province (Grant No. 201300311400), National Natural Science Foundation of China (Grant No. 52179015 and 51779093), Science and Technology Project of Guizhou Province Water Resources Department (Grant No. KT202001), and Innovation Ability Improvement Project in North China University of Water Resources and Electric Power (Grant No. YK-2021-45).

**Data Availability Statement:** Not applicable.

**Acknowledgments:** Thanks for the help provided by Haibo Yang in data collection.

**Conflicts of Interest:** The authors declare no conflict of interest.

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
