# Peer review of "Spatial–Temporal Patterns and Propagation Dynamics of Ecological Drought in the North China Plain"

_water, doi:10.3390/w14101542_

Round 1

Reviewer 1 Report

This work requires a Major Revision. Kindly find the attached pdf with comments for your perusal.

Thanks and sincerely

Reviewer 2 Report

Review report of manuscript titled “Spatial-temporal Patterns and Propagation Dynamics of Ecological Drought in the North China Plain”. Overall, the manuscript is written well, and the study is quiet novel. However, the following revisions need to be carried out before the acceptance of the manuscript.

  1. The introduction section is incomplete. The authors have identified and defined the gaps in the research but missed to state the objectives of the current study. Refer to line no: 89-92. The authors need to state the objectives framed to fill the research gaps identified in the above paragraph.
  2. The literature review focusing on spatio-temporal drought pattern modeling seems weak. Hence, additional literature needs to be reviewed and included in the Introduction section. The authors can include the following latest literatures in their review.

Kwon, M., Kwon, H. H., & Han, D. (2019). Spatio‐temporal drought patterns of multiple drought indices based on precipitation and soil moisture: A case study in South Korea. International Journal of Climatology, 39(12), 4669-4687.

Diaz, V., Perez, G. A. C., Van Lanen, H. A., Solomatine, D., & Varouchakis, E. A. (2020). An approach to characterise spatio-temporal drought dynamics. Advances in Water Resources, 137, 103512.

Bahmani, S., Naganna, S. R., Ghorbani, M. A., Shahabi, M., Asadi, E., & Shahid, S. (2021). Geographically Weighted Regression Hybridized with Kriging Model for Delineation of Drought-Prone Areas. Environmental Modeling & Assessment, 26(5), 803-821.

Li, H., Kaufmann, H., & Xu, G. (2022). Modeling Spatio-temporal Drought Events Based on Multi-temporal, Multi-source Remote Sensing Data Calibrated by Soil Humidity. Chinese Geographical Science, 32(1), 127-141.

Afshar, M. H., Bulut, B., Duzenli, E., Amjad, M., & Yilmaz, M. T. (2022). Global spatiotemporal consistency between meteorological and soil moisture drought indices. Agricultural and Forest Meteorology, 316, 108848.

  1. The Materials and Methodology section is incomplete. The authors just give the details of various methods used and nowhere mention how they are implemented for their spatio-temporal analysis of ecological drought. The main logic behind the appropriateness of the models for the spatial data analysis should be made. I think the number of stations and the spread of the area brings a big question on the validity of the assumptions which needs to be properly explained.
  2. In figure 2, how the number of IMF’s are determined?
  3. The authors need to state limitations of the study if any or drawbacks of the methods used in the conclusion section. Also present the future scope.
  4. How can one predict the ecological drought of various scales in advance? Suggest suitable methods if any.

Reviewer 3 Report

This study investigated the spatial and temporal patterns and propagation dynamics of ecological drought. The temporal evolution and spatial patterns of ecological drought were identified, and propagation times could be estimated. The main issues which I think need to be clarified and worked are presented as follows :

#1. Various studies have been conducted to analyze the relationship between vegetation and climate, and in particular, studies to estimate propagation time have been performed. Compared with related studies, the innovativeness of this study should be emphasized. Also, what are the new viewpoints into propagation dynamics mentioned in the abstract?

#2. Information on the NDVI dataset is unclear. Giving information on the temporal and spatial resolution of the NDVI dataset used will be helpful in understanding this paper.

#3. Overall, the description of the methodology in this study is lacking. A description of the methodology may be provided as a reference, but it should describe where or how the methodology is applied in this study. For example, the purpose of applying ESMD is unknown in the methodology of the present paper.

#4. There are many methodologies presented in this study. Similar to comment #3, each method is explained very simply. Therefore, the purpose of applying each methodology should be clearly stated.

#5. Lines 269-270: However, looking for studies analyzing the relationship between vegetation and climate (especially for China), it has been reported that the propagation time from a meteorological drought to a vegetation drought is long (Fang et al., 2019). Delayed time of vegetation may vary depending on different climatic characteristics and watershed conditions. In future studies, it is recommended to apply SPEI of more diverse timescales.

Fang, W., Huang, S., Huang, Q., Huang, G., Wang, H., Leng, G., ... & Guo, Y. (2019). Probabilistic assessment of remote sensing-based terrestrial vegetation vulnerability to drought stress of the Loess Plateau in China. Remote Sensing of Environment, 232, 111290.

#6. I believe that to estimate the propagation time from meteorological drought to ecological drought, the study area should be analyzed on a grid scale of satellite data, rather than divided into sub-zones. As shown in Figure 1, since various altitudes and land cover types appear within one sub-zone, the propagation time is expected to be very different for each grid.

Reviewer 4 Report

The manuscript presents results of research on the spatial and temporal patterns and propagation of ecological drought recorded in the North China Plain in the multi-annual period 1999-2019. In my opinion, the paper is interesting, especially when taking into account the growing problem of droughts in the northern part of China. I would recommend the manuscript to be published in the journal. However, the Authors should make some improvements prior to its final acceptance for publication. Major flaws are as follows:

  1. Figure 1: The figure caption does not correspond to the figures content (elevation and land cover), and lacks reference to the respective sub-figures (1a and 1b). Elevations (sub-figure 1a) should be expressed in meters above sea level (m a.s.l.). In my view, it is not necessary to show the meteorological stations on the two maps, the same with the administrative boundaries shown twice in the figure legends. Please correct.
  2. Table 1: What is the meaning of the table caption (”The NCP in this study”)? It is not informative and needs to be modified.
  3. Please supplement the description of the study area with information on its climate, hydrography and hydrology.
  4. Page 4: “2.2.3. Teleconnection Factors” – please add information on the teleconnections source data.
  5. I would suggest merging the sub-chapter “Dataset” into the chapter “Materials and Methodologies”.
  6. Figure 2: “Decomposition of VCI from 1999 to 2019 in the NCP based on ESMD”. The figure caption (title) does not fully correspond to the sub-figures content. Please be more specific and correct it.
  7. Table 2 needs to be rearranged, as its content does not fit in the individual columns of the table.
  8. Figures 6 and 8: please standardize the color scale in these two figures, as in my opinion the increasing and decreasing trends and the percentage of area having these trends should have the same color scale.
  9. Figure 9: it is unclear to me why the correlation coefficients are marked alternately with “R” and “r”. Moreover, I would suggest modifying the color scale by removing the blue color, as the cool colors (including blue) generally represent negative correlations. Additionally, I would suggest deleting dots in the abbreviations of months, i.e.: “Jan” instead of “Jan.”, “Feb” instead of “Feb.” etc.
  10. Figures 10 and 11: please add a label (symbol) describing what the color scale on the right side of the figures shows.
  11. The paper requires some language improvements.

Generally, the paper can be accepted after major revision, in accordance with the aforementioned comments.

Round 2

Reviewer 1 Report

Dear authors,

Thank you for revising your manuscript

Reviewer 4 Report

Explanations and corrections made by the Authors are satisfactory. I recommend to accept the paper for publication in present form.

This manuscript is a resubmission of an earlier submission. The following is a list of the peer review reports and author responses from that submission.